# Maximal Effort Cytoreduction in Epithelial Ovarian Cancer: Perioperative Complications and Survival Outcomes from a Retrospective Cohort

**DOI:** 10.3390/jcm12020622

**Published:** 2023-01-12

**Authors:** Dimitrios Haidopoulos, Vasilios Pergialiotis, Eleftherios Zachariou, Ioakim Sapantzoglou, Nikolaos Thomakos, Emmanouil Stamatakis, Nikolaos Alexakis

**Affiliations:** 1First Department of Obstetrics and Gynecology, Division of Gynecologic Oncology, “Alexandra” General Hospital, National and Kapodistrian University of Athens, 11528 Athens, Greece; 2Department of Anesthesia, “Alexandra” General Hospital, 11528 Athens, Greece; 3First Department of Propedeutic Surgery, National Kapodistrian University of Athens, Hippocration Hospital, 11528 Athens, Greece

**Keywords:** maximal effort, surgical complexity, epithelial, ovarian cancer, survival, morbidity

## Abstract

**Background**: Rates of maximal effort cytoreductive surgery in ovarian cancer patients increase gradually the last decade. The purpose of the present study is to evaluate factors that contribute to survival and morbidity outcomes in this group of patients. **Methods**: We retrospectively reviewed patient records of epithelial ovarian cancer patients with an intermediate and high Mayo Clinic surgical complexity score, operated between January 2010 and December 2018. **Results**: Overall, 107 patients were enrolled in the present study with a median age of 62 years (23–84) and a follow-up of 32 months (2–156). Thirteen Clavien-Dindo grade IIIa complications were documented in 10 patients (9.3%). Of all the investigated factors, only stage IVb (*p* = 0.027) and interval debulking surgery (*p* = 0.042) affected overall survival rates. Overall survival outcomes of patients operated on a primary setting started to differentiate compared to those that received neo-adjuvant chemotherapy after the 4th postoperative year. **Conclusions**: Maximal effort cytoreductive procedures should be considered feasible in the modern surgical era, as they are accompanied by acceptable rates of perioperative morbidity. Hence, every effort should be made to perform them in the primary setting, rather than following neoadjuvant chemotherapy as current evidence favor increased survival rates of patients that will likely surpass an interval of observation of more than 4 years.

## 1. Introduction

Epithelial ovarian cancer ranks fifth in cancer-related deaths among women and is the fourth-most common gynecologic malignancy. The lifetime risk of ovarian cancer is 1.1% with an estimated 10.6 new cases and 6.5 deaths per 100,000 women annually [1]. Surgical treatment combined with chemotherapy remains the cornerstone for treatment of these cases [2,3]. The last decade the introduction of targeted therapies, including anti-VEGF targeting drugs and PARP-inhibitors, has significantly increased the survival of ovarian cancer patients [4,5]. However, even with optimal cytoreduction and strict adherence to the chemotherapy schemes, mortality rates remain extremely high, as the majority of ovarian cancer cases is diagnosed with advanced stage disease due to the absence of specific symptoms and signs during the early stages of the disease [6].

Optimal cytoreduction is associated with survival outcomes and every effort should be made to reduce tumor residual to non-visible disease (R0 excision) at completion of the operation [7,8]. This is not, however, always feasible as advanced stage disease may be surgically unresectable, especially when the bowel mesentery is involved or in the presence of extensive miliary disease involving the small intestine as even the presence of millimetric disease seems to significantly affect survival rates [9]. A potential strong predictor of disease-specific survival seems to be the timing of surgery as it is hypothesized that patients treated with primary debulking have better survival outcomes compared to those that undergo neoadjuvant chemotherapy followed by interval debulking surgery. However, results from currently available randomized trials do not show a clear benefit of upfront surgery [10,11]. The ENGOT ov33/AGO-OVAR OP7 TRUST trial will help determine whether cases with FIGO stage IIIB-IV ultimately benefit from primary cytoreductive surgery [12]; however, its results are expected to be available in the second half of 2023.

Maximal effort cytoreduction has been proposed as a feasible surgical technique by several researchers [13,14]; however, it should be always performed in centers with appropriate expertise in the peri-operative of ovarian cancer patients [15]. Recently, in a population-based study conducted by the Dutch Gynecological Oncology Group researchers observed that by increasing the surgical aggressiveness perioperative morbidity increases substantially leading to potential elongation of the interval to adjuvant chemotherapy, therefore reducing the total survival benefit these patients receive during surgery [16]. Therefore, optimization of the process of selection of potential candidates is of paramount importance, and this, according to current evidence, seems to be feasible [17].

In the present study we report the feasibility of intermediate and high complexity score procedures in terms of perioperative morbidity of FIGO stage IIIb-IV ovarian cancer patients as well as parameters that influence survival outcomes.

## 2. Materials and Methods

### 2.1. Study Design

The study was based on a retrospective chart review of records of all patients who underwent maximal effort cytoreduction for advanced ovarian cancer between January 2010 and December 2018. We predefined as maximal effort cytoreduction all procedures that were characterized by a moderate and high Mayo Clinic surgical complexity score [18]. This scoring system assigns one point of surgical complexity for each of the following: hysterectomy and bilateral salpingo-oophorectomy, omentectomy, pelvic lymphadenectomy, paraortic lymphadenectomy, pelvic peritoneal stripping, abdominal peritoneal stripping and small bowel resection. Large bowel resection, diaphragmatic stripping/resection, splenectomy, and liver resection are assigned two points of surgical complexity. Finally, rectosigmoidectomy with reanastomosis is assigned 3 points of surgical complexity. A sum of ≤3 points indicates a low complexity score, 4–7 points an intermediate complexity score and ≥8 points a high complexity score. All cases undergoing maximal effort cytoreduction were primarily discussed in the multidisciplinary team meeting of our hospital, which involves the participation of gynecologic oncologists, medical oncologists, radiologists, radiotherapists, and gynecologic oncology specialized pathologists. Cases are considered for upfront surgery when resectability is deemed possible (no extra-abdominal metastases or extensive liver parenchymal metastases that would require excision in an extent that would exceed that of sphenoid resection) and when patient performance status is considered adequate (Eastern Cooperative Oncology Group (ECOG) status ≤ 3). The intraoperative mapping of ovarian cancer spread system was used to evaluate tumor dissemination as previously described [13]. All operation were performed by three ESGO (European Society of Gynecologic Oncology) accredited gynecologists. The institutional review board of our hospital approved this study prior to its onset (IRB approval number: 631/2020).

### 2.2. Definitions

We classified perioperative complications using the Clavien-Dindo score into major (Grade ≥ IIIa) and minor (Grade I–II) peri-operative complications that occurred within a timeframe of 30 days from the operation. Major peri-operative complications were sub-categorized to those that were directly related to the surgical technique (including bleeding, anastomotic leakage, peritoneal abscess, and wound dehiscence) and those that were indirectly related to the operation (bowel obstruction, pulmonary complications, heart failure, heart failure and myocardial infarct, sepsis and deep vein thrombosis, and pulmonary embolism). As minor peri-operative complications we included cases with lower urinary tract infections, obstructive uropathy, bladder sensory loss, lymphocyst formation, and fever exceeding 38 °C.

Tumor extent was evaluated with the “Intraoperative Mapping of Ovarian Cancer” tool that assesses the presence of cancer after compartmentalizing the abdomen in nine sections [19]. Residual disease at the end of the procedure was classified as non-visible (R0 excision), visible—less than 1 cm (R1 excision) and visible—more than 1 cm (R2 excision).

Progression free and overall survival rates were calculated from the onset of the operation until clinically or radiological evidence disease relapse and until patient death, respectively. Survival outcomes were retrieved from patient records and for cases that were not assessed during the last 30 days from the onset of data retrieval of this information was performed by phone call.

### 2.3. Statistical Analysis

Statistical analysis was performed using the SPSS 20.0 program (IBM Corp., Armonk, NY, USA). Evaluation of the normality of distributions was performed with graphical methods and the Kolmogorov–Smirnoff analysis. The differences of continuous variables were assessed using the Mann–Whitney and Kruskal–Wallis test (due to the abnormal distribution that was observed during the evaluation of normality) whereas dichotomous variables were analyzed with the chi-square test. Fisher’s exact test was applied wherever the number of observations was lower than five in the case of dichotomous variables. Cox regression analysis (enter method) was carried out in order to assess the independent effect of the stage of disease, tumor histology (serous vs. other), surgical complexity (intermediate/high), presence of residual disease at the end of the operation and of the timing of the procedure (primary mode of treatment or for recurrent disease/the latter being considered as reference variable) on patient survival rates. The Kaplan-Meier method was carried out to perform survival-analyses. The level of significance for all analyses was set to *p* < 0.05.

## 3. Results

### 3.1. Patient and Tumor Characteristics

Overall, 107 patients were enrolled in the present study with a median age of 62 years (23–84). Patients > 60 years old accounted for the majority of cases (59%). Of those, 15 patients (14%) were aged > 75 years, five of them being octogenarians. The performance status of patients was evaluated as good (ECOG 0–2) in the majority of cases (82%).

Most patients were classified as stage IIIc (71%) (Table 1) and accounted for high grade serous carcinoma (76%). In 21 patients unilateral pleural effusion was identified and in nine patients we observed bilateral pleural effusion. Most of the patients underwent upfront debulking surgery (81%), whereas 14 patients underwent interval debulking surgery following three cycles of platinum-based chemotherapy and six patients underwent interval debulking surgery following 4–6 cycles.

### 3.2. Tumor Dissemination, Extent of Resectability and Perioperative Complications

The median extent of tumor dissemination at laparotomy, following assessment with IMO tool, was 8 (4–9). Most of them (73%) had more than six abdominal compartments infiltrated by metastases. No evidence of remaining disease was achieved in 80 women (75% of cases) whereas in 18.7% of cases a R1 excision was performed. Finally, in six cases a R2 excision was deemed necessary.

Two perioperative deaths within a timeframe of 38 days occurred in our series. Both patients were ranked as ECOG status 2. Both of them were the result of peritonitis following gastrointestinal leakage. In the first case the leak was observed in the greater gastric curvature and the second in the colorectal anastomosis that was performed following sigmoidectomy.

Perioperative surgical morbidity was observed in 42 cases. Twelve ECOG status 3 patients developed postoperative surgical morbidity, indicating a significant effect of the performance status on the postoperative course (*p* = 0.0186). The most frequent complication was gastroparesis that persisted for >15 days, which was encountered in 17 patients. In their majority these patients had radical omentectomy combined with excision of the lesser omentum. Treatment consisted of correction of electrolytes, parenteral nutrition, metoclopramide and erythromycin. None of these patients required surgical treatment (ex gastrostomy, jejunostomy). Fever was encountered in 21 cases and was attributed to urinary tract infections (11 cases), pulmonary complications (infection/atelectasis) (eight cases) and anastomotic leak (three cases). Thirteen Clavien-Dindo grade IIIa complications were observed in 10 patients, namely nine high-output pleural effusions that developed following diaphragmatic stripping and required insertion of a Bullau catheter and 4 wound dehiscence, which were treated with wound debridement and re-approximation. Grade ≥ IIIb Clavien-Dindo complications were observed in 12 cases. Of those, three cases had an ECOG performance status 3. Compared to patients with better performance status no differences were observed in rates of severe postoperative morbidity requiring reoperation (*p* = 0.486). Bowel leakage was the most frequent complication encountered in seven cases. All of them were handled surgically with revision of the procedure and formation of a stoma. Hartmann’s procedure was performed in three cases, re-anastomosis of the remaining bowel with loop ileostomy in another three cases, and re-anastomosis of the small bowel with loop jejunostomy in one case. Ileal pouch leak was observed in one case that required re-operation with formation of terminal ileostomy. Wound dehiscence and evisceration was observed in another case that also required reoperation. Peritoneal abscess formation was observed in two cases, one of them requiring revision of the operation for abdominal lavage.

### 3.3. Survival Rates

The median follow-up of our cohort was 32 months (2–156). The recurrence free survival of the entire cohort was 18 months (12.69, 23.31) and the overall survival 47.34 months (36.92, 57.75). Of all the investigated factors, only stage IVb (*p* = 0.027) and interval debulking surgery (*p* = 0.042) significantly affected overall survival rates. Of note, overall survival rates in the interval debulking surgery group start to differentiate compared to the primary debulking group after the 50th month postoperatively (*log-rank* = 0.038) (Figure 1). After subgrouping cases according to the level of tumor resection, we observed that optimally cytoreduced patients had increased survival rates compared to patients with R1 excision (28.91 (20.74, 37.08) months vs. 19.71 (12.99, 26.42) months); however, the result did not reach the appropriate significance level (*log-rank* = 0.389). Neither of the investigated parameters had a detrimental impact on progression-free survival. Table 2 and Figure 2 summarize survival outcomes among subgroups of patients that were selected after considering factors that could potentially affect survival rates.

## 4. Discussion

Our study denotes that maximal effort cytoreductive surgery is feasible in advanced ovarian cancer. Ideally, it should be performed in a primary setting, rather than following neoadjuvant chemotherapy as patients with prolonged survival tend to have better overall survival outcomes (differences started to appear in patients with an overall survival that exceeded 4 years from initial treatment). The actual reason behind this observation remains unclear; however, we can speculate that patients with extremely advanced ovarian cancer that require maximal effort cytoreduction tend to have smaller life expentancy; therefore, the ones that might benefit from primary surgery are those that are most responsive to chemotherapy.

It is important to stress that this observation appears for the first time in the international literature and it may partially explain the absence of a clear survival benefit that is indicated by previous studies such as the SCORPION trial which did not show superiority of primary debulking surgery compared to interval debulking procedures [10]. Similar to the findings of our study, the Kaplan–Meier curves of this trial indicate a potential differentiation in the overall survival of enrolled patients, in favor of primary debulking, after the completion of the 4th year of enrollment, despite the fact that the log-rank result was not significant. In another trial conducted by the Japanese Gynecologic Oncology Group, researchers observed that the non-inferiority of neo-adjuvant chemotherapy was not confirmed, however, the sample size that was required to extract safe conclusions was not reached; hence, the study was regarded as underpowered [20]. Kaplan–Meier curves in this study indicated comparable overall survival rates in a long term follow-up; however, the rates of optimal debulking surgery remained low both for PDS (12% of cases) as well as for IDS operations (31% of cases). In our study, the appropriate sample size was not reached to permit safe evaluation of the impact of timing of surgery in appropriately cytoreduced patients, however, descriptive statistics indicated increased intervals of overall survival the PDS group of patients. This observation has been previously confirmed by Sorensen et al. in a large cohort retrieved by the Danish Gynecological Cancer Database [21]; however, even in this population-based study, the power to retrieve robust conclusions was not reached in the subgroup of appropriate cytoreduced patients.

The actual impact of cytoreductive surgery on survival outcomes of patients with epithelial ovarian cancer seems to be highly dependent.

It should be noted that advanced age (>70 years) should not be regarded as an adverse peri-operative factor as the findings of our study support the feasibility of the technique with comparable survival outcomes. Our study results are in line to those of a previous study that supported the use of ultra-radical resection with acceptable peri-operative morbidity rates in ovarian cancer patients [22]. On the other hand, frailty, indicated in our series by an ECOG performance status 3, seems to be a factor that affects both the extent of resectability as well the rates of postoperative complications.

Perioperative morbidity has been proven to be directly associated with the extent of resection in ovarian cancer patients [23]. In our series we observed that the rate of severe complications (Clavien-Dindo grade 3b or more) was somewhat lower compared to those of previous reports [13,18], indicating that maximal effort cytoreduction is both feasible and ethically acceptable in a wide range of age groups. Several reasons may contribute to this observation. For instance, even slight deviations in surgical complexity scores among published studies can have a wide impact on the actual perioperative morbidity as progressively increasing multiorgan excision has a significant impact on morbidity outcomes [24]. Furthermore, low anastomotic leak rates reported in our series may be the result of increased rates of stoma formation in our institution as previously reported [25]. It should be noted, however, that all cases of death that were recorded in our series of patients were attributed to gastrointestinal leak which occurred despite the involvement of a general surgeon in the anastomosis at primary surgery and reoperation.

Optimal perioperative patient management represents a detrimental step for the management and close monitoring of laboratory and clinical parameters is of crucial importance [26]. In our institution all patients that undergo maximal effort cytoreduction are postoperatively observed in a dedicated surgical high dependency unit with close follow-up of fluid intake and optimization of albumin concentration. Enhanced recovery protocols are widely adopted in order to help patient mobilization and a dedicated physiotherapist is involved in cases at increased risk of respiratory morbidity, emphasizing in patients that had diaphragmatic stripping or resection.

One could speculate that maximal effort debulking surgery would have a negative impact on the quality of life (QoL) of enrolled patients. In our series we were not able to collect this outcome as QoL is not routinely recorded in patient records and given the retrospective nature of our study such data could not be retrieved. Evidence supports, however, that despite an initial small to moderate decrease in physical, role, and emotional function at 6 weeks post-operatively, patients undergoing extensive surgery have comparable QoL at 6–12 months [27].

### 4.1. Strengths and Limitations of Our Study

The main strength of our study is the large follow-up duration of patients undergoing maximal effort cytoreduction, rendering it one of the few published in this field in an international setting. On the other hand, despite the relatively large number of patients enrolled in our series of patients, the appropriate sample size to reach robust conclusions concerning the impact of optimal timing of surgery (PDS vs. IDS) in the subgroup of optimally cytoreduced patients was not reached; hence, this parameter remains to be investigated by larger cohorts. Moreover, given the retrospective design of our study, selection bias cannot be ruled off as several factors may affect surgeon’s decision to proceed with certain types of treatment, including primary debulking surgery over neoadjuvant chemotherapy and maximal effort cytoreduction with optimized tumor resectability. Nevertheless, given the fact that we used a consecutive cohort, we believe that the findings of our study can be considered as pragmatic, reflecting results from the implementation of current guidelines of the European Society of Gynaecological Oncology for the management of patients with advanced epithelial ovarian cancer. A parameter that we did not investigate was the actual extent of cytoreductive procedures performed in our institution compared to those of previous time periods; however, this was predesigned in our study, as we observed that patients operated prior to 2010 were more frequently referred to neoadjuvant chemotherapy; hence, limiting the number of patients receiving high complexity score procedures. Another potential factor that might affect morbidity and survival outcomes is the presence of concurrent diseases, including diabetes, heart, and renal failure. In our series we opted to exclude these factors and chose to use the ECOG performance status to indicate patient frailty in order facilitate the analysis of this cohort, which is limited by the number of enrolled patients. Larger series may help in the future overcome this barrier and evaluate the impact of selected comorbidities on patient outcomes.

### 4.2. Implications for Future Research

Large multi-institutional prospective trials are needed to gain further experience on the impact of selected variables that influence survival rates, including extent of tumor resection, patient age, and adoption of enhanced recovery protocols. Information concerning patients aged >70 years old undergoing high complexity score procedures is of high importance, as this group refers to approximately 30% of cases. The impact of preoperative prehabilitation, particularly in frail patients, needs further research to define the variables which should be targeted to optimize the perioperative outcome and concurrently the survival rates. Finally, we believe that when addressing survival outcomes, an extended follow-up period is needed to evaluate the actual impact of high complexity score procedures as it seems that patients undergoing primary debulking and surviving >50 months have improved overall survival.

## 5. Conclusions

Maximal effort cytoreductive procedures should be considered feasible in the modern surgical era as perioperative major surgical morbidity and mortality rates range within acceptable and comparable rates to those reported in the international literature; however, it should be stressed that these outcomes are based in the experience of referral centers with broad experience in the handling of such cases. Patient frailty seems to be a factor that may affect extent of resectability as well as postoperative morbidity; therefore, the decision to proceed in cases requiring extensive excision should be taken with caution.

## Figures and Tables

**Figure 1 jcm-12-00622-f001:**
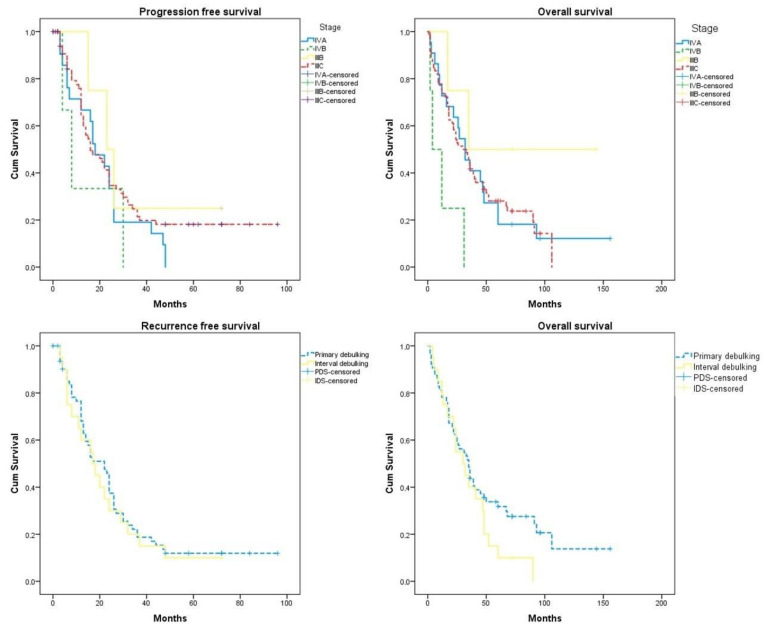
Progression-free survival and overall survival according to the stage of the disease and timing of surgery (primary debulking vs. interval debulking). Differences in overall survival rates start to differentiate from the 50th month (*log-rank* = 0.036).

**Figure 2 jcm-12-00622-f002:**
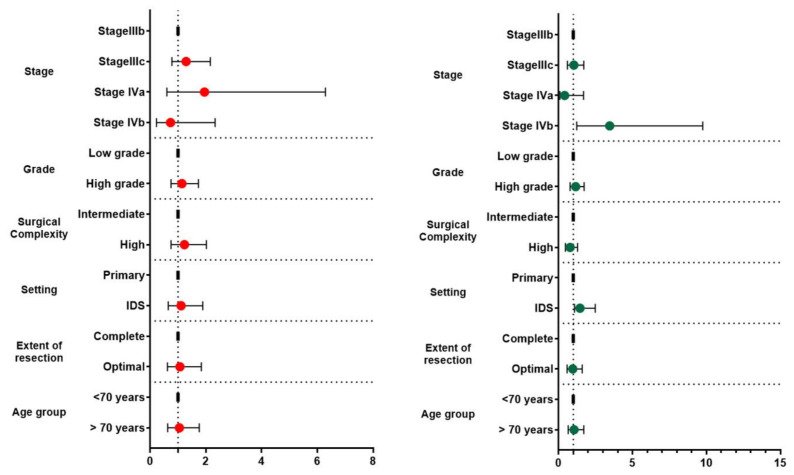
Forest plots of hazard ratios and 95% CIs of factors affecting progression-free (right side—red dots) and overall survival (left side—green dots) rates.

**Table 1 jcm-12-00622-t001:** Patient characteristics at enrolment and peri-operatively.

Age	62 (23–84)
Stage	
IIIb	4
IIIc	76
IVa	23
IVb	4
Histological grade	
Low	11
High	96
Histological type	
Serous	90
Mucinous	2
Clear cell	3
Endometrioid	3
Carcinosarcoma	9
Interval debulking	20
Primary debulking	87
Ascites	
None	24
<500 mL	40
>500 mL	42
ECOG 0/1	88
ECOG 2/3	19
Hysterectomy	106
Omentectomy	105
Pelvic lymphadenectomy	78
Paraortic lymphadenectomy	35
Large bowel resection	31
Large bowel stoma	16
Small bowel resection	10
Small bowel stoma	12
Splenectomy	26
Cholocystectomy	3
Sphenoid liver resection	9
Diaphragmatic stripping	34
Diaphragmatic resection	18
Extent of debulking	
R0	81
R1	20
R2	6

**Table 2 jcm-12-00622-t002:** Survival outcomes among selected subgroups of patients. Progression-free survival and overall survival in months. PFS = progression-free survival, OS = overall survival.

Parameter	Progression Free Survival (95% CI)	Overall Survival Months (95% CI)	Hazard Ratio 95% CI PFS	Hazard Ratio 95% CI OS
Stage				
IIIb	34.00 (12.14, 55.86)	85.00 (26.85, 143.15)	Ref	Ref
IIIc	31.21 (23.19, 39.23)	42.21 (33.55, 50.87)	1.29 (0.78, 2.16)	1.04 (0.60, 1.71)
IVa	21.05 (14.70, 27.39)	46.95 (27.29, 66.15)	1.95 (0.60, 6.29)	0.41 (0.10, 1.69)
IVb	14.00 (2.00, 29.84)	12.25 (2.00, 25.21	0.73 (0.23, 2.33)	3.46 (1.23, 9.75)
Grade of differentiation				
Low grade	26.25 (11.94, 40.56)	48.20 (26.96, 69.44)	Ref	Ref
High grade	25.77 (20.12, 31.41)	46.82 (35.44, 58.20)	1.14 (0.75, 1.73)	1.16 (0.78, 1.73)
Surgical complexity				
Intermediate	31.59 (23.59, 39.59)	42.93 (31.04, 54.82)	Ref	Ref
High	24.74 (15.60, 33.88)	46.51 (34.16, 58.86)	1.23 (0.75, 2.02)	0.78 (0.47, 1.29)
Setting of operation				
Primary	23.15 (14.40, 31.90)	53.75 (39.98, 67.52)	Ref	Ref
Interval	28.13 (21.15, 35.11)	35.15 (24.21, 46.09)	1.11 (0.65, 1.89)	1.45 (1.08, 2.48)
Level of resection				
R0	28.99 (22.17, 35.81)	46.14 (35.23, 57.05)	Ref	Ref
R1	24.91 (15.21, 34.61)	47.92 (27.85, 67.98)	1.07 (0.62, 1.84)	0.96 (0.58, 1.60)
Patient age				
<70 years	26.94 (21.22, 32.66)	45.39 (34.89, 55.89)	Ref	Ref
>70 years	29.48 (16.24, 42.72)	45.60 (26.16, 56.04)	1.05 (0.63, 1.76)	1.05 (0.65, 1.71)

## Data Availability

Data available upon reasonable request.

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
