# Peer review of "Maximal Effort Cytoreduction in Epithelial Ovarian Cancer: Perioperative Complications and Survival Outcomes from a Retrospective Cohort"

_jcm, 2023, doi:10.3390/jcm12020622_

Round 1

Reviewer 1 Report

Thank you for the opportunity to review the manuscript (jcm-2155081) entitled Maximal effort cytoreduction in epithelial ovarian cancer: morbidity and survival outcomes from a retrospective cohort by Haidopoulos et al. the manuscript evaluates the factors that contribute to survival and morbidity outcomes in a certain group of patients via retrospective reviewing of patient records of epithelial ovarian cancer patients with an intraoperative intermediate and high Mayo Clinic surgical complexity score, operated between 2010 and 2018. This manuscript is generally well written. However Strengths and limitations of our study? Comparison with other time periods should be considered, What should be focus on in future research?

Author Response

Thank you for the opportunity to review the manuscript (jcm-2155081) entitled Maximal effort cytoreduction in epithelial ovarian cancer: morbidity and survival outcomes from a retrospective cohort by Haidopoulos et al. the manuscript evaluates the factors that contribute to survival and morbidity outcomes in a certain group of patients via retrospective reviewing of patient records of epithelial ovarian cancer patients with an intraoperative intermediate and high Mayo Clinic surgical complexity score, operated between 2010 and 2018.

This manuscript is generally well written.

However Strengths and limitations of our study? Comparison with other time periods should be considered, What should be focus on in future research?

Authors reply: We thank the reviewer for the kind comment. In the present revision the strengths and limitations section was further expanded (Lines 332-346) and the implications for future research introduced in the conclusion section (Lines 347-358). We did not focus on previous time periods as the extent of debulking surgery was already known to be moderate to minimal in cases involving the upper abdomen and these cases were usually referred for neoadjuvant chemotherapy. A comment for this was added in the present revision (Lines 335-340).

Reviewer 2 Report

This retrospective cohort study explored the effect of maximal effort cytoreduction in epithelial ovarian cancer and provided clinical evidence in the performance of maximal effort cytoreduction on epithelial ovarian cancer patients, but there are some problems to be corrected.

1. The title is " Maximal effort cytoreduction in epithelial ovarian cancer: morbidity and survival outcomes from a retrospective". what are "morbidity and survival outcomes" means? only progress free survival (PFS) and overall survival (OS) were explored.

2. abstract: line 17. what "intraoperative intermediate" means?

3. abstract: line 18. you should provide the detail month of "2010-2018".

4. abstract: line 22. "survival outcomes" refer PFS or OS?

5. conclusion and abstract: line 24. the conclusion "Maximal effort cytoreductive procedures should be considered feasible in the modern surgical era" cannot be proved by this study as this study did not compare the  effect of Maximal effort cytoreductive and other surgery procedures.

6. methods: how the PFS and OS were defined and how the follow-up of this study was performed?

7. methods: line 72: I suggest to describ "Mayo Clinic surgical complexity score" in detail. 

8.  in abstract "results" part, the authors write " Thirteen Clavien-Dindo grade IIIa complications were documented ", but in methods "definitions" part, complications were divided into major and minor. they should be consistent.

9. The results were too long, some important data should be shown in Table and Figure. the statement should be shorter.

10. Table 2 can be shown in forest figure.

11. Auxiliary reference line should be added in Figure 1. I cannot found the survival difference in 4th year at present.

12. The conclusion part should be shorter. only important conclusion should be shown in this part. and some conclusion were not consistent with the results, it must be more precious. 

Author Response

Dear reviewer,

We appreciate your comments and thank you for reviewing our study and considering it for potential publication. In the present revision you will find a point-by-point answer to all requested revisions. We hope that the present version will meet the standards for publication. However, we are willing to further revise if modifications/clarifications are needed.

This retrospective cohort study explored the effect of maximal effort cytoreduction in epithelial ovarian cancer and provided clinical evidence in the performance of maximal effort cytoreduction on epithelial ovarian cancer patients, but there are some problems to be corrected.

  1. The title is " Maximal effort cytoreduction in epithelial ovarian cancer: morbidity and survival outcomes from a retrospective". what are "morbidity and survival outcomes" means? only progress free survival (PFS) and overall survival (OS) were explored.

Authors reply: in our article we provided a descriptive analysis of perioperative surgical morbidity which was observed in 42 cases; hence we chose to include morbidity in our title. A modification of the title was performed, hoping that this will meet the reviewer`s demand. However, if the reviewer finds this inappropriate we may include only survival outcomes.

  1. abstract: line 17. what "intraoperative intermediate" means?

Authors reply: we thank you for this comment. “intraoperative” was erased in the present revision. (Line 36)

  1. abstract: line 18. you should provide the detail month of "2010-2018".

Authors reply: the months are introduced in the present revision (Line 37 and 94-95).

  1. abstract: line 22. "survival outcomes" refer PFS or OS?

Authors reply: we thank the reviewer for this remark. Only OS was significantly affected by the timing of the operation (PDS vs IDS). This was revised in the abstract section as well as the main document (Line 42).

  1. conclusion and abstract: line 24. the conclusion "Maximal effort cytoreductive procedures should be considered feasible in the modern surgical era" cannot be proved by this study as this study did not compare the effect of Maximal effort cytoreductive and other surgery procedures.

Authors reply: the feasibility of the technique was commented on the basis of the relatively low rates of severe morbidity (9.3% of cases). This is specified in the present revision. We hope that the reviewer will accept this comment (Lines 46-47).

  1. methods: how the PFS and OS were defined and how the follow-up of this study was performed?

Authors reply: we thank the  reviewer for this remark. A field was introduced in the methods section to clarify these issues (Lines 133-137).

  1. methods: line 72: I suggest to describe "Mayo Clinic surgical complexity score" in detail. 

Authors reply; we discussed the surgical complexity score in full detail in the present revision (Lines 97-104).

  1. in abstract "results" part, the authors write " Thirteen Clavien-Dindo grade IIIa complications were documented ", but in methods "definitions" part, complications were divided into major and minor. they should be consistent.

Authors: this point is clarified in the present revision (Line 119).

  1. The results were too long, some important data should be shown in Table and Figure. the statement should be shorter.

Authors reply: half of the data presented in the results section referred to information already shown in Table 1 and were removed from the present revision.

  1. Table 2 can be shown in forest figure.

Authors reply: a forest figure was introduced in the present revision. However, we would like to ask from the reviewer to accept Table 2 as well which presents the accurate summary effect and confidence intervals as we believe that this information will help future researchers compare their results to those of our study.

  1. Auxiliary reference line should be added in Figure 1. I cannot find the survival difference in 4th year at present.

Authors reply: significant differences are noted in the overall survival rates of patients undergoing PDS/IDS. This is noted in the present revision (Line 246 and Lines 511-512).

  1. The conclusion part should be shorter. only important conclusion should be shown in this part. and some conclusion were not consistent with the results, it must be more precious. 

Authors reply: several information were moved to a newly formed section named implications for future research and the conclusion was toned down to indicate that the feasibility relies in the relatively acceptable rates of perioperative morbidity, noting, however, that caution should be exerted in frail patients.

Reviewer 3 Report

This manuscript by Haidopoulos et al. is an interesting manuscript describing cytoreductive surgery in ovarian cancer patients. The author has seen an increase in the survival rate of advanced stages cancer patients undergoing cytoreductive surgery. However, there are some important questions that need to be answered.

MAJOR COMMENTS:

1. Cancer is often followed by secondary diseases, like diabetes. How were secondary diseases controlled before carrying out the cytoreductive surgery?

2. Cytoreduction targets the affected normal cells. How does it affect the cancer stem cells?

3. It is interesting to see survival rates. However, it is not significant. Can the author propose some therapies along with cytoreduction in the conclusion?

4. In Table 1, please mention the grades for histological type cancer, like if serous = 90, how many were grade III or IV?

Author Response

Dear reviewer,

We appreciate your comments and thank you for reviewing our study and considering it for potential publication. In the present revision you will find a point-by-point answer to all requested revisions. We hope that the present version will meet the standards for publication. However, we are willing to further revise if modifications/clarifications are needed.

This manuscript by Haidopoulos et al. is an interesting manuscript describing cytoreductive surgery in ovarian cancer patients. The author has seen an increase in the survival rate of advanced stages cancer patients undergoing cytoreductive surgery. However, there are some important questions that need to be answered.

MAJOR COMMENTS:

  1. Cancer is often followed by secondary diseases, like diabetes. How were secondary diseases controlled before carrying out the cytoreductive surgery?

Authors reply: we thank the reviewer for this remark. Patient status was controlled only with the ECOG classification in the present study. Indeed, chronic diseases may affect the perioperative outcome of high complexity procedures. However, this parameter was not evaluated in the present study due to the significant heterogeneity of chronic diseases in the cohort which would result in several small subgroups. This point is discussed in the limitation section. (Lines 340-346)

  1. Cytoreduction targets the affected normal cells. How does it affect the cancer stem cells?

Authors reply: we thank the reviewer for this remark. It is known that surgical cytoreduction is of special importance on survival rates of epithelial ovarian cancer patients; however, its impact on cancer stem cells remains, to date, although there are reports that suggest that residual cancer stem cells may be responsible for poor treatment outcomes in this population of patients. The aim, however, of our study was not to evaluate the presence of stem cells; hence, we do not believe that a reference to cancer stem cells would be relevant to our findings and would rather confuse readers.

  1. It is interesting to see survival rates. However, it is not significant. Can the author propose some therapies along with cytoreduction in the conclusion?

Authors reply: proposing other potential therapies, including targeted therapy (including PARP and anti-VEGF therapy) is beyond the purpose of this study as it refers mainly to the impact of surgical debulking on survival outcomes. HIPEC therapy may be an option, however, to date it has not been accepted by international societies; hence, its discussion seems to be non-relevant to our findings as none of the cases has been treated with HIPEC in the present series.

  1. In Table 1, please mention the grades for histological type cancer, like if serous = 90, how many were grade III or IV?

Authors reply: grading of tumors has been dichotomized to low and high grade according to the FIGO classification and in order to help evaluate survival outcomes of this cohort.

Round 2

Reviewer 2 Report

Thanks for authors to revise this manuscript carefully. I think this revised manuscript can be published.

Reviewer 3 Report

The authors have sincerely answered all the concerned queries. The manuscript is greatly enhanced and can be considered for publication.